# Sound Source Localization Indoors Based on Two-Level Reference Points Matching

**Shuopeng Wang** [1],*[ID], **Peng Yang** [2,3] **and Hao Sun** [2,3][ID]

1   School of Information Engineering, Tianjin University of Commerce, Tianjin 300134, China
2   School of Artificial Intelligence, Hebei University of Technology, Tianjin 300130, China
3   Engineering Research Center of Ministry of Education for Intelligent Rehabilitation, Tianjin 300130, China
*   Correspondence: wangsp87921@hotmail.com; Tel.: +86-183-2260-1354

**Abstract:** A dense sample point layout is the conventional approach to ensure the positioning accuracy for fingerprint-based sound source localization (SSL) indoors. However, mass reference point (RPs) matching of online phases may greatly reduce positioning efficiency. In response to this compelling problem, a two-level matching strategy is adopted to shrink the adjacent RPs searching scope. In the first-level matching process, two different methods are adopted to shrink the search scope of the online phase in a simple scene and a complex scene. According to the global range of high similarity between adjacent samples in a simple scene, a greedy search method is adopted for fast searching of the sub-database that contains the adjacent RPs. Simultaneously, in accordance with the specific local areas' range of high similarity between adjacent samples in a complex scene, the clustering method is used for database partitioning, and the RPs search scope can be compressed by sub-database matching. Experimental results show that the two-level RPs matching strategy can effectively improve the RPs matching efficiency for the two different typical indoor scenes on the premise of ensuring the positioning accuracy.

**Keywords:** fingerprint-based sound source localization; two-level matching strategy; adjacent reference point searching; greedy search method; clustering method

## 1. Introduction

Sound source localization (SSL) has received significant research attention in the field of audio signal processing, and it is widely used in intelligent robots, blind spot detection and underwater detection [1–3]. What is more, microphone array SSL is a spatial spectrum estimation problem for broadband short-time stationary signals, the research results of which can also be used in mobile communication, sonar detection and radar detection.

Usually, traditional SSL methods can be divided into three categories: high-resolution spectral estimation method [4], steered beamforming method [5] and time delay of arrival (TDOA) method [6]. These methods can transform the spatial geometric relationship between the sound source and the microphone array into a spatial spectrum, spatial beam and TDOA, respectively, first and then work out the location of the sound source accordingly. Due to the low computational complexity and hardware cost, the TDOA SSL method is widely used in sound source location and tracking [7,8]. As a parametric positioning method, the TDOA SSL method usually uses the space geometrical propagation model to obtain the position of sound source [9–14]. In practice, the signal propagation model should be simplified as follows:

(1)   The sound source is a particle without size and shape.
(2)   The signal propagates in a homogeneous space.
(3)   The sound signal is omnidirectional.

The SSL methods based on a geometry model can achieve ideal results outdoors, where the actual signal propagation model is similar to the idealized simplification model

explained above. However, due to the complexity of the indoor environment, the ideal signal propagation model may be altered by the multipath effect, shadowing effect, fading effect and delay distortion caused by walls, floors, furniture and ceilings [15,16]. Meanwhile, it is difficult to provide compensation for model distortion analytically due to the high complexity of sound field characteristics indoors [17,18].

As a non-parametric localization method, fingerprint-based localization can locate the target point by the matching between the real-time signal and the database that contains the historical location information of the service area. This method can take full advantage of the similarity of signal characteristics of adjacent samples in the service area and effectively reduce the location error caused by the modelling error and measurement error in the geometric model method indoors. Compared with the precise measurement requirements and the stringent restraint of the application scenario for the parameter positioning method, avoiding sharp changes in the positioning environment, as the only requirement for the fingerprint-based SSL method, it is much easier to be satisfied in practical applications [19,20].

As the basis of fingerprint-based SSL indoors, the positioning database scale directly affects the positional accuracy of the SSL system [21,22]. In practical applications, in order to make the location fingerprint database better reflect the distribution characteristics of the sound field, it is usually necessary to arrange a large number of sampling points in the location service area. However, the matching calculation for searching adjacent RPs from the large-scale database will greatly reduce the online positioning efficiency. Therefore, the fingerprint-based SSL encountered difficulties in applications with high real-time requirements such as mobile robot auditory positioning, indoor abnormal sound source positioning and speaker positioning [23].

In order to improve the efficiency of fingerprint-based SSL indoors, various methods are proposed to optimize the offline sampling process and the online positioning process. For the offline sampling phase, Khalajmehrabadi et al. [24] adopted the sparse database recovery method based on interpolation to reduce the initial RPs to improve the efficiency of offline sampling. An interpolation is a mathematical tool for estimating the unknown function value using available function values of other variables. Interpolation methods for scattered data are widely implemented in mathematical, industrial and manufacturing applications. Radial basis function (RBF) [25], linear [26], inverse distance weighting (IDW) [27] and kriging [28] are well-known interpolation methods for positioning database expansion. Due to the initial RP reduction, the interpolation methods can effectively increase the collection efficiency in the fingerprint database [29]. However, since the virtual RPs generated by the interpolation method still needed to participate in the adjacent RPs matching, the interpolation method cannot obviously improve the efficiency of the online positioning phase of the fingerprint-based SSL indoors.

Selective matching of the target point and the RPs can reduce the computation amount of the online positioning procedure. Many studies consider dividing the database into many sub-databases, and then selecting the sub-database that is most likely to contain the adjacent RPs to reduce the computation amount for matching RPs [30]. Study [31] introduces a variety of database partition methods based on coordinate grid division, which can effectively improve the efficiency and stability of the fingerprint-based localization method. Liu et al. [32] proposed a minimum enclosure method to realize the flexible definition of the grid size in the coordinate grid division method. However, the coordinate partitioning method may be affected by the subjective judgment of the operator, which may lead to problems such as inconsistent database partitioning results and high positioning errors caused by the mismatching of adjacent RPs.

According to the complexity of sound field characteristics, indoor position scenes can be divided into simple and complex scenes. For a simple scene, the problem of adjacent RPs searching can be regarded as a spatial distance optimization problem that satisfies the optimal substructure. The local search algorithm is a kind of general algorithm that can solve global optimization problems through a series of local optimization processes. The

greedy search algorithm is a simple and efficient local search algorithm that can improve search efficiency by avoiding the exhaustive exercises usually needed to find the optimal solution. For complex indoor scenes, cluster analysis can automatically divide the different RPs into the same sub-databases where samples have high similarity. Compared with the coordinate partitioning method, the feature clustering partitioning method is more consistent with the distributed rule of adjacent RPs [33].

In this paper, we deal with the issue of improving the localization efficiency of the fingerprint-based SSL method. A two-level RPs matching strategy is proposed in this paper to improve the search rate for the adjacent RPs. In the first-level matching process, two methods are adopted to shrink the adjacent RPs search scope. For simple scenes, a greedy search strategy is adopted for fast searching of the sub-database that contains the adjacent RPs, and for complex scenes, the search scope can be compressed by sub-databases matching based on the database partition by clustering method. The performance of the proposed algorithms is evaluated by comparing them with the traditional linear RPs matching method, and the practical experiment results verify the effectiveness of the proposed method.

The rest of the paper is organized as follows: In Section 2, the general process of fingerprinting acoustic localization is briefly introduced. In Section 3, the two-stage RPs matching method is stated to improve the efficiency of SSL. In the first level search, the greedy algorithm and the Fuzzy c-means clustering algorithm are proposed separately to shrink the RPs search range of the second level search in the two different scenes indoors. Section 4 presents the implementation details and evaluates the performance of the novel methods from the results obtained. Finally, some conclusions are drawn in Section 5.

## 2. Fingerprint-Based SSL

As shown in Figure 1, the process of fingerprint-based SSL consists of two phases: the offline sampling phase for database construction and the online positioning phase for vocal target location estimation.

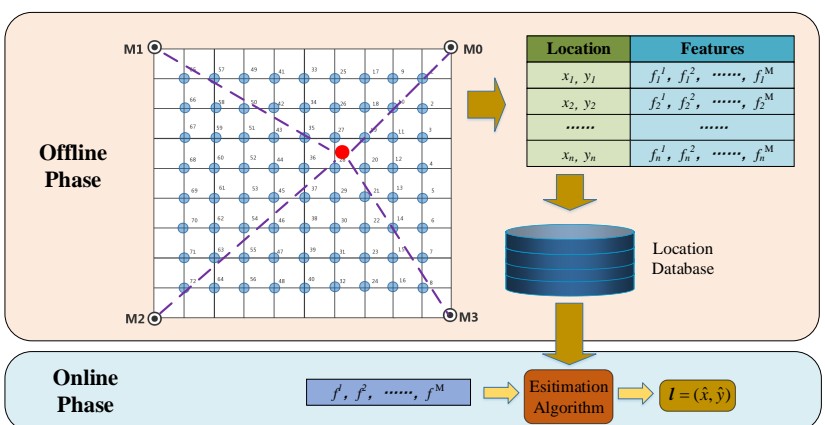

**Figure 1.** Illustration of fingerprint-based SSL process.

### 2.1. Offline Sampling Phase

Generally, the offline phase includes three steps. First, the coordinates of the samples are determined according to the environment and precise requirements of the positioning service area. Then, the positioning signal is released at each sampling location and received by the sound source positioning system with four microphones, M1, M2, M3 and M4, as shown in Figure 1. Finally, the RPs are made up of the coordinates of samples and the corresponding location features extracted from the positioning signal. The RPs are also known as position fingerprints:

$$S_n = [L_n, F_n]^T, \quad n = 1, 2, \ldots, N. \tag{1}$$

where $S_n$ corresponds to the fingerprint collected at the $n$th sampling point, and $N$ is the total quantity of the sampling point in the positioning service area; $L_n=[x_n, y_n]$ and $F_n = [f_n^1, f_n^2, \cdots, f_n^M]$ mean the coordinates and the feature vectors of the $n$th RPs. $M$ is the total number of the positioning features and $f_n^m$ means the $m$th feature in feature vector $F_n$.

Sound intensity, frequency spectrum and time difference of arrival (TDOA) are closely related features of sound source position [34–37]. Among them, TDOA is widely used in real-time positioning for the characteristics of the low computational complexity and a small amount of data [38]. In this work, there are four microphones in the system, and we choose TDOA as the positioning feature. Thus $F_n = [\Delta t_{n1}, \Delta t_{n2}, \Delta t_{n3}]$. $\Delta t_{ni}$ represents the TDOA value of the received signal between the reference microphone and the other three microphones at the $i$th reference point. We collect the fingerprint at each sampling point and establish the positioning database defined as follows:

$$D=[S_1, S_2, , \cdots, S_N] \tag{2}$$

*2.2. Online Positioning Phase*

When the sound signals of the auditory target are observed by the SSL system, the feature vector of the observed signal will be extracted and matched with each sample in the positioning database. Then, the target position can be calculated by the estimation algorithm through the adjacent RPs from the RPs matching process. Exactly the same estimation algorithm is used in the RADAR system, the weighted k-nearest neighbour (WKNN). Algorithm [39] is used for the SSL process in this paper:

$$l = \sum_{i=1}^{k} \omega_i L_i \tag{3}$$

where $l = (\hat{x}, \hat{y})$ is the positioning result of the auditory target, $k$ is the number of adjacent RPs and $L_i = (x_i, y_i)$ is the coordinates of the $i$th adjacent RPs. The according weight $\omega_i$ of the $i$th adjacent RPs can be calculated through the inverse distance weighting method as follow:

$$\omega_i = \frac{1/(dis_i + \varepsilon)}{\sum\limits_{j=1}^{k} 1/(dis_j + \varepsilon)} \tag{4}$$

where $dis_i$ represents the Euclidean distance between the target point and the $i$th adjacent RP in feature space. $\varepsilon$ is a small random value for avoiding the denominator from being 0 (the $dis_i$ may be 0 when the target point is very close to a certain sample point).

## 3. Two-Level Matching Method for Adjacent RPs Searching

Usually, empty rooms, halls, corridors or other scenes with open space or few obstacles can be considered to be the typical simple scene for SSL. At the same time, home and office environments, where the positioning space may be separated into relatively independent regions by obstacles such as furniture and walls, etc., can be considered to be the complex scene.

The sound field characteristics of simple and complex scenes are both analysed through the pairwise correlation of the RPs in the positioning service area. As shown in Figure 2(a1), 72 RPs are uniformly distributed in the square positioning area without obstacles, and as Figure 2(a2) shows, the correlation coefficient of each of the two RPs obviously decreases with the increase in the distance. Take the ratio of correlation coefficient beyond 0.6 into consideration, and as Figure 2(a3) shows, when the distance is within 1.5 m, most cross-correlation values of two RPs are relatively high. However, when the distance is more than 1.5 m, the ratio of the correlation value beyond 0.6 will decrease rapidly along with increases in the distance. In the global scope of the simple scene, the correlation between the RPs is relatively high in the small scope area and will decrease rapidly along with the increase in the distance.

As shown in Figure 2(b1), 64 RPs are distributed in the square positioning area with 4 desks. According to Figure 2(b2), when the physical distance of the RPs is within 1.5 m, most of the according correlation coefficient can still reach 0.6, which is better than the simple scene with the same distance. However, as Figure 2(b3) shows, when the physical distance is beyond 2 m, the ratio of the correlation value beyond 0.6 shows obvious fluctuations. This is because the correlation coefficient within each sub-positioning service area separated by the physical plane becomes stronger, and the correlation coefficient between different locations becomes weaker.

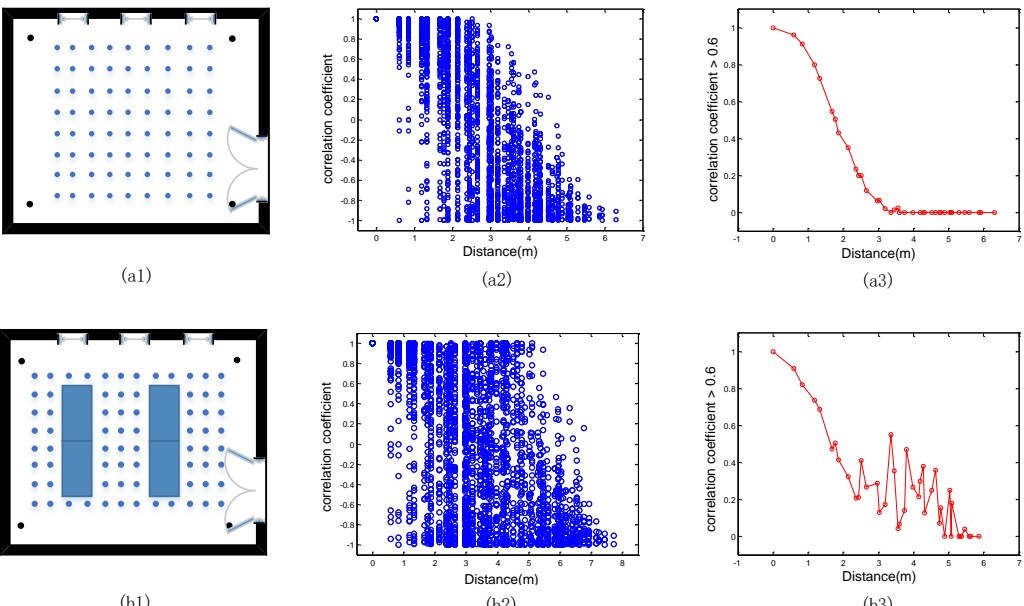

**Figure 2.** The correlation coefficient value between RPs in two typical indoor positioning scenes: (**a1**) setting of experimental environment for simple indoor positioning scene; (**a2**) relationship between correlation coefficient and distance of RPs in simple indoor positioning scene; (**a3**) the ratio of the correlation value for RPs beyond 0.6 in simple indoor positioning scene; (**b1**) setting of experimental environment for complex indoor positioning scene; (**b2**) relationship between correlation coefficient and distance of RPs in complex indoor positioning scene; (**b3**) the ratio of the correlation value for RPs beyond 0.6 in complex indoor positioning scene.

*3.1. Adjacent Subset Searching Based on Greedy Algorithm*

Greedy algorithm refers to choosing the best or most optimized option in each step so as to bring about the best or optimized overall performance of the algorithm [40]. For instance, in the problem of adjacent RPs searching, if the nearest RPs are chosen as the searching center for each searching step, it can be regarded as a kind of greedy algorithm. A greedy algorithm is particularly effective in solving the problem of the optimal substructure. Optimal substructure means that the local optimum can determine the global optimum. Put simply, the problem can be divided into sub-problems for a solution. The optimum for the sub-problems can recur to the optimum for the final problem.

Adjacent RPs of the database refers to the RPs that are closest to the target point in the feature space. Therefore, adjacent RP searching is a global optimization problem of spatial distance essentially. As Figure 2 stated previously in Section 3, the TDOA characteristics of samples are of high local correlation in a global range, and the correlation value will rapidly decrease when the physical distance increases. Therefore, the search of adjacent RPs basically meets the greedy rule of optimal substructure in the simple indoor scene. As shown in Figure 3, the matching process of adjacent RPs based on a greedy algorithm is composed of three parts.

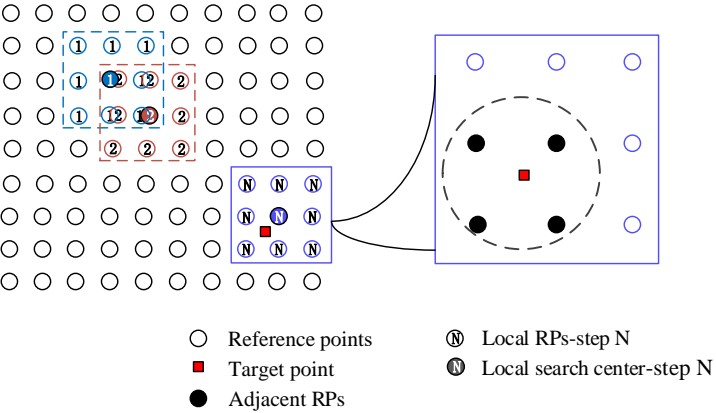

○ Reference points     Ⓝ Local RPs-step N

■ Target point     ◑ Local search center-step N

● Adjacent RPs

**Figure 3.** The RPs matching process based on a greedy algorithm.

First, the Euclidean distance between the RPs and the target point in the feature space is selected as the objective function $f(i)$, and an RP is randomly selected from the location database $\boldsymbol{D} = [\boldsymbol{S}_1, \boldsymbol{S}_2, \cdots, \boldsymbol{S}_N]^T$ is appointed to be the first search center (i.e., the initial optimal solution). Then, other solutions in the neighborhood of the optimal solution (i.e., the RPs near the search center) and the optimal solution itself constitute the current local search database $\boldsymbol{D}^l$:

$$\boldsymbol{D}^l = [\boldsymbol{S}_1^l, \boldsymbol{S}_2^l, \cdots, \boldsymbol{S}_g^l]^T \tag{5}$$

where $\boldsymbol{S}_i^l = [\boldsymbol{L}_i^l, \boldsymbol{F}_i^l]$ $i = 1, 2, \ldots, g$. and $g$ is the total number of samples in a locally searched subset. We then calculate the objective function value of each solution in the local search database $\boldsymbol{D}^l$:

$$f(i) = \left\| \boldsymbol{F} - \boldsymbol{F}_i^l \right\|_2, \ i = 1, 2, \ldots, g. \tag{6}$$

where $\boldsymbol{F}$ is the feature vector of the positioning target and $\boldsymbol{F}_i^l$ is the feature vector of the $i$th element in the local search subset. We select the solution that minimizes the value of the objective function as the new optimal solution:

$$\boldsymbol{c} = \boldsymbol{S}_{\arg\min_{i \in \{1,2,\ldots,g\}} f(i)} \tag{7}$$

If the optimal solution no longer changes, the greedy search process will end. The optimal solution of the current local search process will be the globally optimal solution (i.e., the nearest RPs of the target point). At the same time, the current local search subset $\boldsymbol{D}^l$ will be the adjacent subset; otherwise, continue to repeat the search process of Equations (5)–(7). At last, in the adjacent subset $\boldsymbol{D}^l$, according to the distance $dis_j$ between the target point $\boldsymbol{F}$ and each RPs $\boldsymbol{F}_j^l$ of subset $\boldsymbol{D}^l$, select the adjacent RPs group $\boldsymbol{D}_a$ for position estimation:

$$\boldsymbol{D}_a = \boldsymbol{D}_a \cup \boldsymbol{S}_{\arg\min_{j \in \{1,2,\ldots,n_c\}} dis_j} \tag{8}$$

### 3.2. Adjacent Subset Searching Based on Clustering Method

The clustering method can classify datasets according to the similarity between samples and classify new sampling points. For fingerprint-based SSL, the clustering method can be used to separate the positioning database into several sub-databases and classify the pending target into a corresponding category. Then, the RPs matching range will be reduced compared to the global linear matching method. The process of fingerprint-based SSL using the clustering method is shown in Figure 4 in brief.

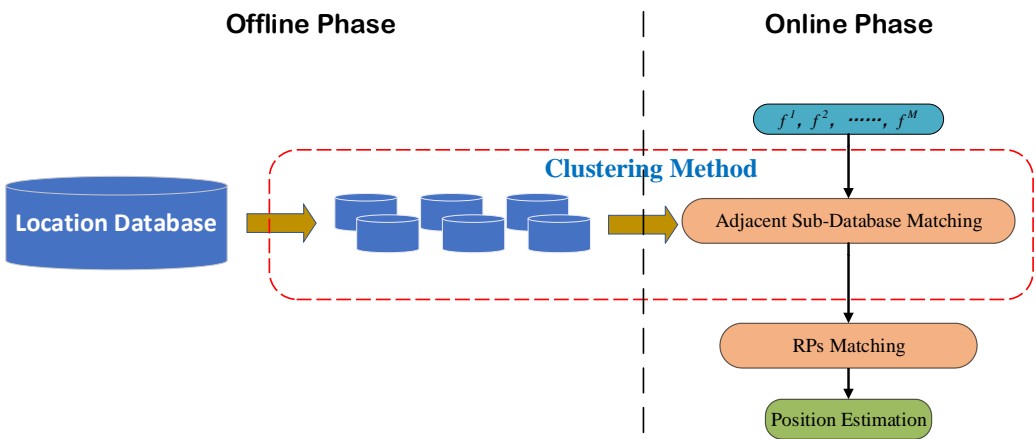

**Figure 4.** The RPs matching process based on clustering analysis.

In many cases, it is difficult to classify the targets reasonably by the hard clustering method such as the K-means algorithm, which was adopted in our previous work [41], because the relationship between RPs in practice is vague and uncertain. In this case, the soft clustering method can more scientifically and reasonably divide the database. As a typical clustering method, the Fuzzy c-means algorithm can use fuzzy mathematics to analyze the uncertainty of the sample properties, and the clustering partition will be completed according to the membership degree of samples. For the RPs matching process in the fingerprint-based SSL, the first-level searching (i.e., adjacent sub-database searching) process based on the Fuzzy c-means method can be shown as follows:

Step 1: determine the number of sub-databases $k$, which means the positioning database will be divided into $k$ clusters.

Step 2: assign a membership degree to each cluster for each RP, which meets the following conditions:

$$\sum_{c=1}^{k} u_{cj} = 1, \quad 0 \le u_{cj} \le 1, \quad c = 1, 2, \dots, k. \quad j = 1, 2, \dots, N. \tag{9}$$

where $u_{cj}$ represents the membership degree of the $RP_j$ to cluster $c$, and the value is defined between 0 and 1 (when the value is 1, the RP is exclusive to the cluster, $c$), and $N$ is the total number of RPs.

Step 3: calculate the clustering center and update the membership matrix of RPs. Specifically, the objective function of the Fuzzy c-means algorithm is:

$$J = \sum_{c=1}^{k} \sum_{j=1}^{N} (u_{cj})^{\gamma} D_j^c \tag{10}$$

where $D_j^c = \left\| F_{center}^c - F_j \right\|$ represents the distance between the clustering center of cluster $c$ and $RP_j$. The calculation method is the same as (6). $\gamma$ is the weighted index, and its value range is $[1, \infty)$. In order to minimize the objective function, the Lagrange multiplier method can be used to construct the function:

$$\mathrm{F}(U, \Phi, \lambda) = J + \sum_{j=1}^{N} \lambda_j (\sum_{c=1}^{k} u_{cj} - 1) = \sum_{c=1}^{k} \sum_{j=1}^{N} (u_{cj})^{\gamma} D_j^c + \sum_{j=1}^{N} \lambda_j (\sum_{c=1}^{k} u_{cj} - 1) \tag{11}$$

where $U = [u_{cj}]$, $\Phi = [F^c_{center}]$, $c = 1, 2, \ldots, k.$ $j = 1, 2, \ldots, N.$, $\lambda_j$, is the lagrange multiplier, and the constraint condition is (9). By differentiating the input parameters, the minimization condition of the objective function can be translated into:

$$F_c = \frac{\sum\limits_{j=1}^{N} (u_{cj})^\gamma F_j}{\sum\limits_{j=1}^{N} (u_{cj})^\gamma} \tag{12}$$

$$u_{cj} = \frac{1}{\sum\limits_{\tau=1}^{k} \left(\dfrac{D_{cj}}{D_{\tau j}}\right)^{2/(\gamma-1)}} \tag{13}$$

Through Formula (12), the new clustering center $U = [u_{cj}]$, $c = 1, 2, \ldots, k.$ $j = 1, 2, \ldots, N.$ can be generated, and then the new membership matrix can be obtained through Formula (13).

Step 4: after the clustering center is generated, we decide whether the result is convergent by the objective function (10): when the condition is not met, return to step 3, and complete the whole updating process through the cyclic iteration of Formulae (12) and (13). When the convergence condition is satisfied, run the next step for clustering results output.

Step 5: output the final clustering center and membership matrix (i.e., the cluster information of the RPs).

## 4. Experimental Results

To demonstrate the performance of the proposed RPs matching method, real-world experiments have been carried out in a practical environment. The room is $9.64 \times 7.04 \times 2.95 \text{ m}^3$, where the noise is about 40 dB and the walls are not insulated. The simple scene and complex scene of the experiments are shown in Figure 5. The positioning area is a rectangular plane with a length of about 6 m and a width of about 5 m. The 4-channel microphone array is composed of the MPA201 microphones produced by the BSWA Technology Co., Ltd., Beijing, China. The microphones are installed at four vertices of the positioning area with a height of about 1.35 m above the floor. The type of the acquisition card is known as NI9215A from NI company. The sampling frequency is set as 100 kHz, and the sampling period is set as 1 s. The sound source is a bluetooth speaker with a height of 0.20 m embedded on a mobile robot. Its shape is approximately cubic, and the sound unit is composed of three identical speakers on three sides. In view of its small size and horizontal symmetry, its directivity is not considered in this paper.

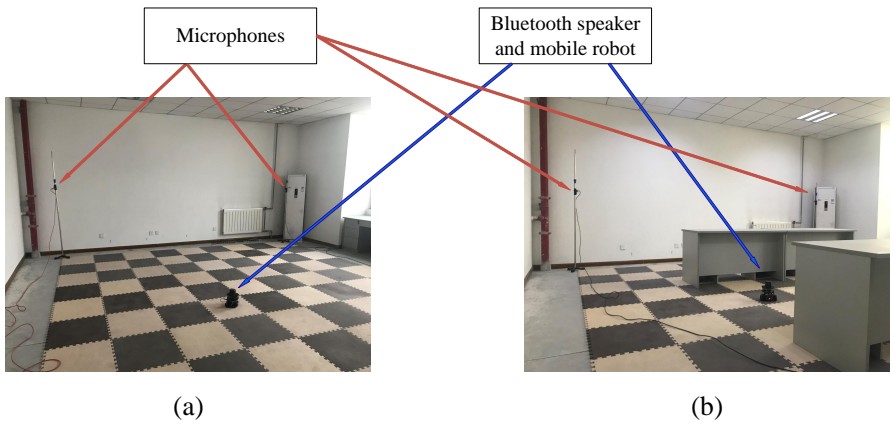

**Figure 5.** The experimental environment: (**a**) simple positioning scene; (**b**) complex positioning scene.

As shown in Figures 5a and 2(a1), in the simple positioning scene, the RPs for the positioning database are uniformly distributed in the location service area by grid division, and

the distance between each RPs is 0.593 m. The total number of the RPs is 72, and there are another 13 test points used for target point estimation. As shown in Figures 5b and 2(b1), in the complex positioning scene, the uniformly distributed RPs in local areas are divided by the obstacles (desks) in the location service area, and the distance between each RPs in a local area is 0.593 m. The total number of RPs is 64, and there are another 18 test points used for target point estimation.

### 4.1. Simple Scene

In order to investigate the effectiveness and stability of the greedy search algorithm in searching the target point's adjacent sub-database, this paper carried out a verification experiment in a simple location scene indoors. The scale of the local search sub-database is set as nine; that is, the current search center (a randomly selected RP of the database) and the eight RPs around it are included in one search process. In the offline sampling stage, the RPs have been sorted by row and column, so the relative positions of the RPs in the database can be directly calculated and compared with the ordinal numbers of its rows and columns. For example, if the RP(a, b) (the RP at row a and column b) is randomly selected as the search center, other RPs of the local search group can be selected as:

$$\left\{ \begin{array}{ccc} RP_{a+1,b-1} & RP_{a+1,b} & RP_{a+1,b+1} \\ RP_{a,b-1} & RP_{a,b} & RP_{a,b+1} \\ RP_{a-1,b-1} & RP_{a-1,b} & RP_{a+1,b+1} \end{array} \right\} \tag{14}$$

The adjacent sub-database search process was independently run 10 times for each test point, and the search results are shown in Table 1. Where, $Step_w$ refers to the maximum number of searching steps beyond the optimal number in 10 independent tests for each test point:

$$Step_w = \text{Max}(steps_i^a - steps_i^o), \quad i = 1, 2, \ldots 10. \tag{15}$$

and $Step_m$ represents the average number of steps beyond the optimal number as:

$$Step_m = \frac{1}{10} \sum_{i=1}^{10} steps_i^a - steps_i^o. \tag{16}$$

As shown in Table 1, the actual searching steps are basically the same as the optimal number for most test points, and the worst value is two steps beyond the optimal number. Even with the initial search center randomly selected, the greedy search algorithm can steadily find the adjacent sub-database that contains the locating target, and the search path is close to the optimal path.

**Table 1.** The relative searching steps of the actual path to the ideal path at each test point.

| No. | $Step_w$ | $Step_m$ |
|---|---|---|
| 1 | 2 | 0.15 |
| 2 | 0 | 0 |
| 3 | 1 | 0.1 |
| 4 | 0 | 0 |
| 5 | 0 | 0 |
| 6 | 0 | 0 |
| 7 | 0 | 0 |
| 8 | 1 | 0.1 |
| 9 | 0 | 0 |
| 10 | 0 | 0 |
| 11 | 1 | 0.2 |
| 12 | 0 | 0 |
| 13 | 0 | 0 |

As an example, the ninth test point was selected to illustrate the stability of the greedy search algorithm. As shown in Figure 6, there are 9 different search paths appearing in the 10 independent searches, among which search path 3 in Figure 6c appeared 2 times. All the search paths successfully completed the search of the adjacent sub-database, and all the search paths except path 7 were optimal paths, but path 7 did not increase the time of the search steps.

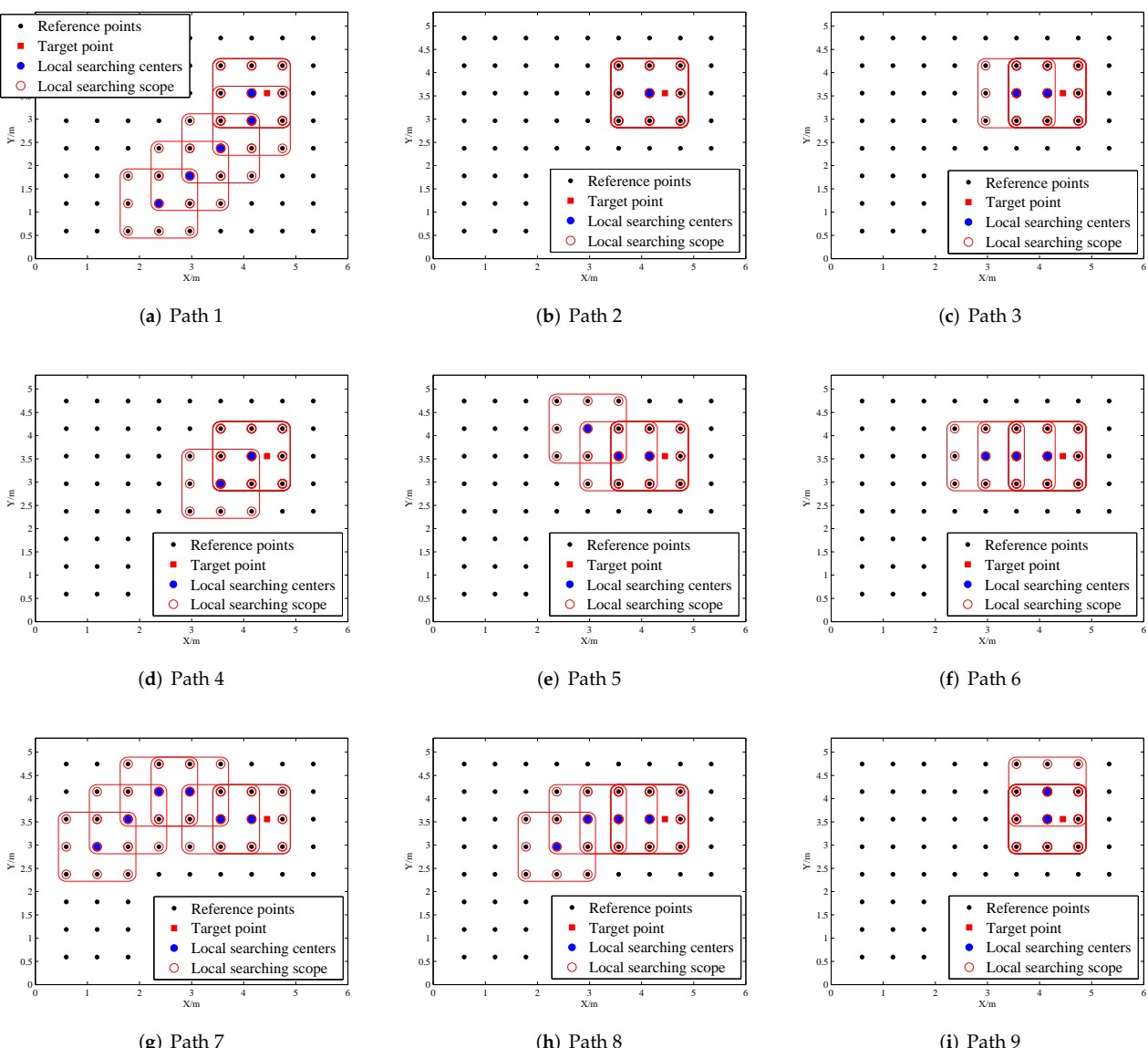

**Figure 6.** The search path of 10 independent tests for the ninth test point.

All of the test points gained equally precise results as the traditional linear matching method in the experiments. What is more, as Figure 7 shows, RP numbers 54, 55, 60 and 62 are selected as the adjacent RPs of test point 9 by the traditional linear matching method, where RP 60 is the mismatched adjacent RP, which results in test point 9 gaining a positioning error of 0.1293 m at last, which is more than the position method based on the greedy RPs searching method. By which the adjacent RPs mismatch phenomenon is avoided, and the positioning error is improved to 0.0377 m.

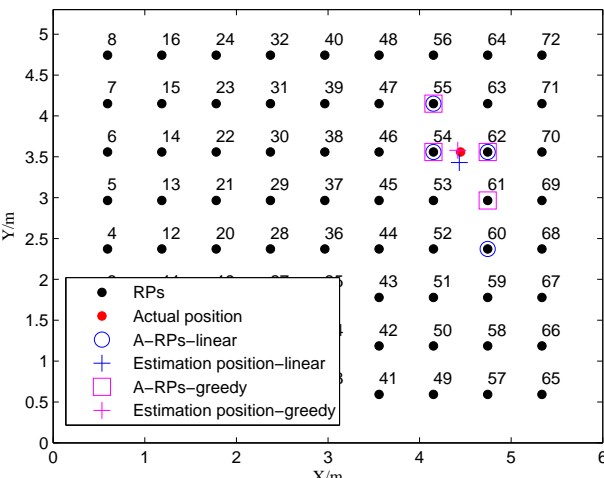

**Figure 7.** Adjacent RP searching results and position estimation results of the ninth test point based on two RP matching methods.

### 4.2. Complex Scene

According to the analysis of the computational complexity in the online location process based on clustering division, the more sub-databases divided into the offline stage, the higher the positioning efficiency in the online process. However, excessive partitioning of the location database may separate the real adjacent RPs of the same target point into different sub-databases, thus affecting the results in the RPs mismatch and reducing the positioning accuracy. The Fuzzy c-means method is used to analyze the positioning results of different clustering number positions. The partitioning results of clustering numbers ranging from 1 to 6 are shown in Figure 8.

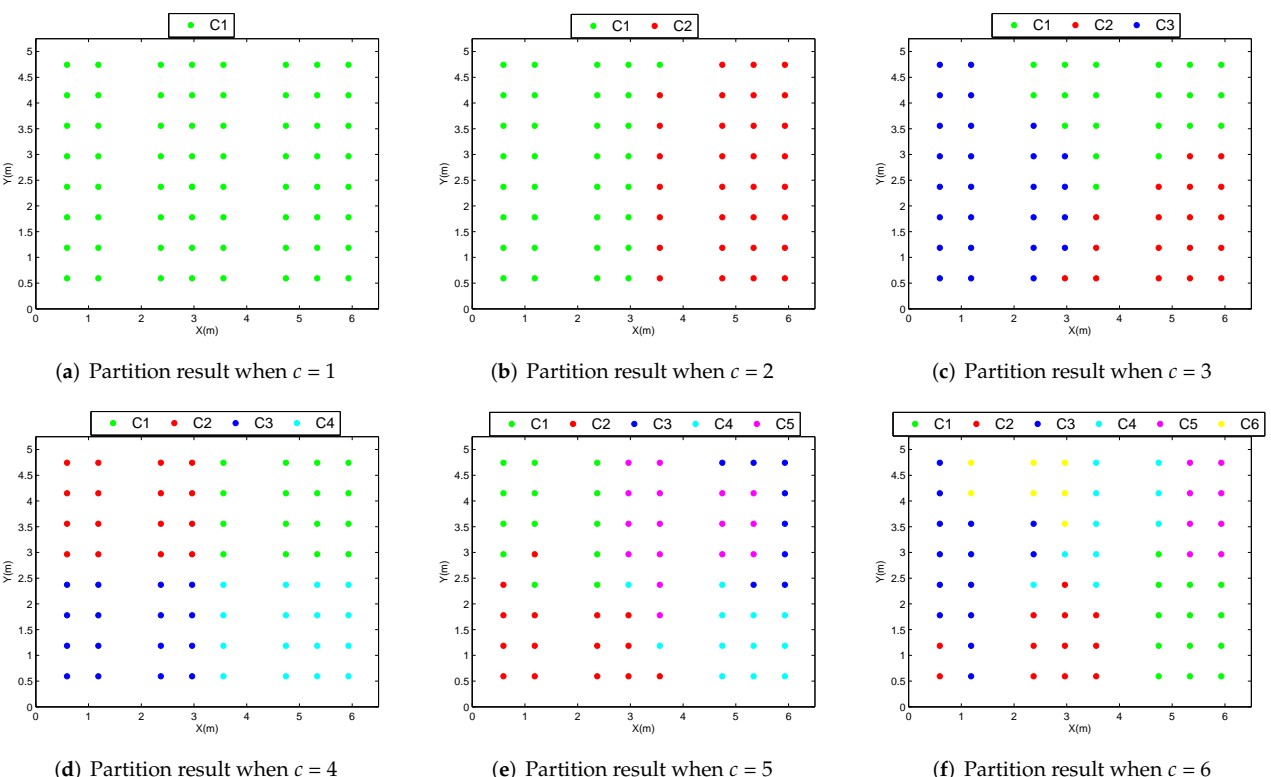

**Figure 8.** The RP partition results of the Fuzzy c-means clustering method in a complex scene.

When the clustering number is set as $c$ = 1, 2, 3 and 4, RPs of the same sub-database are tightly gathered in the connected areas. At the same time, the number of RPs in different sub-databases are basically the same. However, as the partition count increases to 5, as Figure 8e shows, 1 outlier appears in C2 (15 red points), 1 outlier appears in C4 (12 turquoise points) and C3 (8 blue points) is scattered and contains significantly fewer RPs at the same time. When the partition number increased to 6, as Figure 8f shows, 2 outliers appear in C2 (12 red points), 1 outlier appears in C4 (10 turquoise points) and, on the whole, the partition results are obviously imbalanced.

The WKNN position estimation algorithm is adopted for the localization test. As shown in Figure 9, the mean error and maximum error vary with the changes in the number of clusters. Compared with the traditional SSL based on global linear RP matching, the positioning accuracy of SSL based on the clustering analysis is slightly improved when the clustering number is 2, 3 or 4. However, when the clustering number increased to 5, the positioning results began to deteriorate significantly ,and the average error exceeded 0.1850 m, while the maximum error reached 0.7950 m. At the same time, 72.22% of test points cannot meet the positioning accuracy requirement of 0.2000 m.

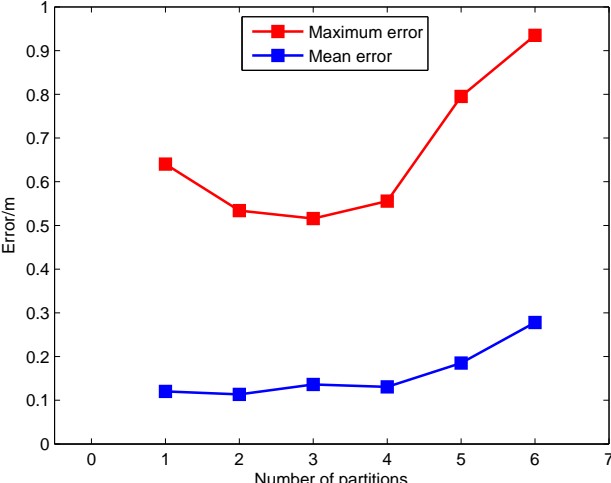

**Figure 9.** The mean error and maximum error of location estimation by different RP partition numbers.

As Table 2 shows, there is a comparison of the traditional case without a partition and four sub-databases cases based on clustering analysis. Where $MQ_a$ means the average matching quantity, $MT_a$ means the average matching time, $E_a$ means the average error and $E_m$ means the maximum error. The $MQ_a$ is reduced by 74.1% through database partition based on clustering analysis, which results in the according reduction of $MT_a$ in SSL based on the two-level RP matching method. In the positioning accuracy comparison, compared with the traditional linear matching method, the $E_a$ and $E_m$ of the two-level matching method based on database partitioning are improved by 13.18% and reduced by 8.47%, respectively. The positioning accuracy between the two RPs matching methods is almost the same.

**Table 2.** Comparison of the localization results between the traditional RPs matching method and the novel method based on clustering analysis.

| Matching Method | $MQ_a$/Times | $MT_a$/s | $E_a$/m | $E_m$/m |
|---|---|---|---|---|
| No partition | 64 | 0.0232 | 0.6402 | 0.1203 |
| 4 sub-databases | 16.6 | 0.0052 | 0.5558 | 0.1305 |

## 5. Conclusions

In this paper, a two-level RP matching strategy is proposed to improve the online positioning efficiency of the fingerprint-based SSL method. In the first level search, the greedy search algorithm and the Fuzzy c-means clustering algorithm are proposed separately to shrink the RP search range of the second level search in the two indoor scenes of different complexities. According to the local similarity in the global range of positioning services in the simple indoors scene, the global optimum task of adjacent database searching is divided into a series of local optimal problems of partial RP matching. The adjacent sub-database is finally obtained through the continuous transfer of the local search center. At the same time, according to local similarity characteristics in some local regions of the positioning services area in the complex indoor scene, the positioning database is divided into a certain number of sub-databases in the offline phase. In the online positioning phase, the matching of the adjacent sub-database can be found for rapid adjacent RPs matching on the promise of ensuring positioning accuracy. In general, the two-level PR matching method can effectively improve the efficiency of SSL and improve the positioning accuracy to a degree. However, the determination of the local search range in the greedy searching algorithm and the clustering number in the database partition needs further study.

**Author Contributions:** S.W. conceived and designed the experiments and wrote the paper; P.Y. and H.S. contributed to project research scheme formulation. All authors contributed to the final version. All authors have read and agreed to the published version of the manuscript.

**Funding:** This research is supported by the National Natural Science Foundation of China (No. 61773151 and 61703135), Hebei Province Natural Science Fund Project (No. F2017202119).

**Institutional Review Board Statement:** Not applicable.

**Informed Consent Statement:** Not applicable.

**Data Availability Statement:** Not applicable.

**Acknowledgments:** The authors thank all the reviewers and editors for their valuable comments and work.

**Conflicts of Interest:** The authors declare no conflict of interest.

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
