# Peer review of "Sound Source Localization Indoors Based on Two-Level Reference Points Matching"

_applsci, doi:10.3390/app12199956_

Round 1
Reviewer 1 Report
Dear Authors,
I have reviewed your paper with interest. I think this paper is well written.
I have some comments and questions. Please answer these.
---------------------------------
1) It may be difficult because there are many figures, but if possible, please adjust the layout and reduce the gaps on the page as much as possible.
2) Can the RP database once created be used as-is for different sound sources?
3) What is the actual processing time required to locate the sound source?
---------------------------------
sincerely yours
Reviewer 2 Report
Authors proposed two-level matching strategy to shrink the adjacent RPs searching scope. Literature review background is appropriate to show the previous work about the related research. The mathematical analysis with experimental results are good. Figure quality looks good. English grammar looks fine. Thus, the manuscript could be minor revision.
1. Data availability section is missing.
2. Please use abbreviated journal names in the reference section.
3. Please provide city, country, and date information of the conference papers in the reference section.
4. Figure 2 labels seems to be a littble it small.
5. In Figure 9, the error was increased a lot. Why is that ? Is there any method to further reduce that ?
6. In Figure 9, how about the results author could be expected if the number of the positions are higher than 10 ?
7. If the positions are located in 3-axis, how the proposed method could be changed ?
